# Research on Dynamic Trajectory Planning Based on Model Predictive Theory for Complex Driving Scenarios

**DOI:** 10.3390/s25237241

**Published:** 2025-11-27

**Authors:** Hongluo Li, Hai Pang, Hongyang Xia, Yongxian Huang, Xiangkun Zeng

**Affiliations:** 1School of Automobile and Transportation Engineering, Guangdong Polytechnic Normal University, Guangzhou 510665, China; melhluo@gpnu.edu.cn (H.L.); xhysummer@gpnu.edu.cn (H.X.); yongxian@mail.ustc.edu.cn (Y.H.); 2School of Innovative Engineering, Macau University of Science and Technology, Macau 999078, China; 2240001005@student.must.edu.mo

**Keywords:** autonomous driving, trajectory planning, model predictive control theory, dynamic driving scenarios

## Abstract

Autonomous driving, a transformative automotive technology, is currently a major research focus. Trajectory planning, one of the three core technologies for realizing autonomous driving, plays a decisive role in the performance of autonomous driving systems. The key challenge lies in planning an optimal trajectory based on real-time environmental information, yet significant research gaps remain, particularly for dynamic driving scenarios. To address this, our study investigates lane-changing trajectory planning in dynamic scenarios based on model predictive control (MPC) theory and proposes a novel dynamic lane-changing trajectory planning method. First, kinematic models for both the host vehicle and surrounding vehicles are established. Then, following the core components of MPC theory, we construct a prediction model, define an objective function, and formulate constraints for the rolling optimization step. Finally, the optimal control sequence derived from the optimization is processed using a least-squares fitting method to generate a lane-changing trajectory that demonstrates real-time adaptability in dynamic environments. The proposed method is validated through simulation studies of three typical driving conditions on a co-simulation platform. The results confirm that the planned trajectory exhibits excellent dynamic real-time adaptability, thereby contributing a foundation for achieving full-scenario autonomous driving.

## 1. Introduction

With the vigorous development of new-generation information technologies such as artificial intelligence, autonomous driving technology based on pure electric vehicles as carriers has become a new strategic direction in the global automotive industry layout. The key technologies of autonomous driving mainly include positioning and perception, decision-making and planning, control and execution, etc. Among these, planning a reasonable and ideal trajectory that meets the requirements of real driving scenarios has become a research difficulty in realizing full-scenario autonomous driving. With the continuous deepening of research on this technology, the trajectory planning technology for autonomous vehicles under simple constraints has gradually matured; however, its development still faces severe challenges. Particularly, trajectory planning in complex dynamic driving scenarios, which requires dynamic randomness and high real-time performance, has become a challenging problem under more complex constraints that urgently needs to be addressed in the trajectory planning of autonomous driving technology.

At present, research on this problem adopts various methods with different characteristics. Among them, curve fitting methods [1,2], artificial potential field methods [3,4,5,6,7,8], sampling-based methods [9,10], and methods based on deep learning and reinforcement learning [11,12,13,14] are widely used in current autonomous driving trajectory planning. However, these methods still suffer from limitations such as high computational complexity and proneness to deadlock. The B-spline curve method is a commonly used approach in curve fitting. Y. Zhang et al. [1] proposed the use of B-spline curves to generate smooth trajectories that satisfy centripetal acceleration constraints, which is suitable for in-lane driving scenarios. B.S. Oldaç et al. [2] attempted to solve the trajectory generation and tracking problem of intelligent vehicles by performing linear interpolation on discrete points generated by B-spline curves. However, the B-spline curve method often has problems such as easy deformation of the overall curve caused by single-point adjustment. As a result, control points need to be recalculated frequently in dynamic scenarios, which affects the real-time performance of trajectory planning. The artificial potential field method guides the robot to avoid obstacles and move toward the target by setting target attractive forces and obstacle repulsive forces. Z. Zhang et al. [3] proposed an algorithm integrating global and local path planning based on the artificial potential field method. By optimizing the gravitational and repulsive force functions acting on the vehicle, this algorithm overcomes the failure problem of the traditional artificial potential field method in specific scenarios. G. Du et al. [6] first conducted systematic modeling and mathematical description of the planning and control problems of autonomous vehicles, and then used the global heuristic planning algorithm based on the artificial potential field method to generate real-time optimal motion sequences. H. Zhang et al. [8] optimized the problems of unreachable target and low success rate in the artificial potential field method by improving the repulsion field function to solve the unreachable target issue and selecting a virtual target point to make the system escape the trap when it falls into one. Although such methods can effectively construct models, they generally have problems of low path search efficiency and an inability to ensure path optimality. Sampling-based methods construct paths by randomly generating samples in the environment. Z. Chen et al. [9] adopted a longitudinal sampling strategy based on preset driving modes, which improved the probabilistic completeness of existing parameterized curve sampling technologies, thereby realizing fast and effective obstacle avoidance. G. Li et al. [10] proposed an improved sampling-based hybrid A* algorithm. First, a hybrid potential field model that characterizes the vehicle’s workspace and integrates traffic rules and obstacle distribution was proposed as a heuristic function, and environmental topology prior information was introduced to generate a set of directional motion primitives, which further improved the feasibility of the planned path. Although such methods can find the optimal path, the speed of approaching the optimal solution is relatively slow.

In addition, learning-based methods have gradually attracted attention, especially the application of deep learning and reinforcement learning. Z. Yu et al. [11] proposed a learning-based Frenet planning network, which samples and selects the most human-like terminal states through learning strategies to generate safe trajectories. Z. Bai et al. [12] proposed an improved deep reinforcement learning path planning algorithm. First, the Automated Mobile Robot (AMR) path planning was modeled as a Markov Decision Process (MDP), and a Double Deep Q-Network (DDQN) was used to obtain the optimal adaptive solution. Finally, the B-spline curve theory was applied to realize path smoothness optimization. J. Nan et al. [13] proposed a longitudinal car-following trajectory planning method based on offline sampling maximum entropy Deep Inverse Reinforcement Learning (DIRL). This method collects natural driving data and uses a deep neural network to replace the linear function of traditional inverse reinforcement learning, which greatly improves the fitting ability to human reward functions. R. Chai et al. [14] specifically focused on the parking maneuver problem of autonomous ground vehicles. In the motion planning stage, a Recurrent Deep Neural Network (RDNN) was innovatively used to approximate the optimal parking trajectory, and the internal correlation between vehicle states was fully explored through the recurrent network structure. In terms of tracking control, a controller based on Adaptive Neural Networks (ALNN) was designed. By dynamically adjusting network parameters, the system stability and error convergence were theoretically guaranteed. Although these methods perform well in handling complex environments and nonlinear problems, they still face challenges such as long training time and large data demand.

In recent years, the Model Predictive Control (MPC) algorithm has garnered significant attention and been widely adopted due to its advantages of good control effect, strong robustness, and convenient handling of various constraints in controlled variables and manipulated variables of the process. Among relevant studies, D. J. Kim et al. [15] proposed a lateral vehicle trajectory planning and control algorithm based on MPC for automatic vertical parking systems. In the research of Bassam Alrifaee [16], vehicle collision avoidance control was studied based on the MPC method, and the effectiveness of this method was verified through simulations and experiments. Zhao et al. [17] designed a two-layer controller for trajectory planning and tracking based on the MPC algorithm and a fuzzy adaptive PID trajectory tracking controller to improve the driving safety of intelligent vehicles. Q. Zeng et al. [18] addressed the problems of insufficient emergency obstacle avoidance capability and vulnerability to noise interference in vehicle collision avoidance systems, and constructed a Cooperative Adaptive Cruise Control (CACC) model integrating multi-constraint MPC. Q. Dai et al. [19] proposed a Model Predictive Decision-Making (MPDM) method integrating lane-changing time constraints based on the MPC method. I. Askari et al. [20] transformed the MPC problem of motion planning into a Bayesian state estimation problem, and then developed a new type of implicit particle filtering/smoothing algorithm. This algorithm is implemented through multiple sets of Unscented Kalman Filters/Smoothers and has the characteristics of high sampling efficiency, fast calculation speed, and excellent estimation accuracy. D. Chu et al. [21] proposed a trajectory planning and tracking framework, which obtains the target trajectory through the artificial potential field method and uses an MPC controller combined with PID feedback to achieve accurate tracking. Experimental and simulation results show that compared with traditional MPC, this framework has significant improvements in tracking accuracy and steering smoothness, especially under curve driving conditions, where its steady-state error can approach zero. G. Yu et al. [22] proposed a bidirectional multi-vehicle collaborative collision avoidance system based on the MPC framework to solve the collision risk prevention and control problem in multi-vehicle head-on driving scenarios. H. Yang et al. [23] proposed a dynamic obstacle avoidance strategy for autonomous vehicles based on MPC by establishing new collision constraints considering road width and vehicle geometric dimensions, ensuring a safe distance between the host vehicle and obstacle vehicles. J. Wang et al. [24] proposed a new obstacle avoidance method based on event-triggered MPC. By setting the trajectory tracking error as the trigger condition, the frequent collaborative operation between the planning layer and the control layer was realized. Y. Wang et al. [25] innovatively considered the motion prediction of other traffic participants and used MPC to optimize the trajectory according to the current state of the vehicle. Through the collaborative optimization strategy of differentiated prediction horizons and coordinate transformation, it not only simplified the integration process of constraints, but also made the planning results more intuitive. However, the aforementioned methods generally have problems such as excessively high computational complexity or insufficient adaptability when facing dynamic and complex working conditions. Therefore, in view of the characteristic that constraint conditions and planning objectives in trajectory planning often change with different scenarios, researching trajectory planning technologies for different scenarios under complex constraints based on control theory with predictive capabilities will effectively solve the key problem of trajectory planning in complex dynamic driving scenarios, thereby accelerating the application and industrialization of autonomous driving technology in engineering.

Based on the current research landscape, this paper conducts relevant research on the challenging issue of trajectory planning in complex dynamic scenarios using the MPC method and proposes a novel trajectory planning approach tailored for complex dynamic driving scenarios. The organization of this paper is as follows. In Section 2, a vehicle kinematic model including the host vehicle and surrounding vehicles is established. In Section 3, the implementation process of dynamic trajectory planning based on model prediction is deduced and analyzed in detail. In Section 4, simulation analyses are carried out under different working conditions based on the built co-simulation platform to verify the effectiveness of the proposed method. Finally, the conclusion and future work are summarized in Section 5.

## 2. Establishment of a Vehicle Kinematic Model

Autonomous driving needs to be realized through the control of the vehicle’s kinematic or dynamic system. Establishing a reasonable vehicle system model is not only a prerequisite for designing a model predictive controller, but also the foundation for achieving vehicle motion planning and control.

### 2.1. Vehicle Kinematic Modeling (Kinematic Model)

Kinematics examines the laws of object motion from a geometric standpoint, encompassing changes in spatial position, velocity, and other parameters over time. Applying kinematic models in vehicle path planning algorithms ensures that the planned paths are practically feasible and satisfy the kinematic geometric constraints during driving; meanwhile, kinematic models can also reflect vehicle motion characteristics with relatively high accuracy.

The vehicle steering motion model is shown in Figure 1 and Figure 2. In the inertial coordinate system XOY, (X_f_, Y_f_) and (X_r_, Y_r_) represent the coordinates of the centers of the vehicle’s front axle and rear axle, respectively. φ denotes the vehicle’s yaw angle (heading angle). The research object is assumed to be a front-wheel-steering vehicle, where δ_f_ is the front-wheel steering angle, v_f_ is the velocity at the center of the front axle, v_r_ is the velocity at the center of the rear axle, and l is the wheelbase. In the variables, the subscript “f” stands for “front”, and “r” stands for “rear”; the same notation applies hereinafter.

Since the research object is a front-wheel-steering vehicle, during the vehicle’s steering movement, the instantaneous center of rotation P lies on the extension line of the rear axle, where R represents the steering radius of the rear wheels. Let M be the center of the vehicle’s rear axle and N be the center of the front axle. It is assumed that the vehicle’s sideslip angle at the center of mass remains constant during steering, meaning the vehicle’s instantaneous steering radius is the same as the road’s curvature radius.

At the rear axle’s travel center (Xr, Yr), the velocity can be expressed as(1)$$v_{r}={\dot{X}}_{r}\cos\varphi +{\dot{Y}}_{r}\sin\varphi$$

Then, the kinematic constraint at the rear axle center M can be expressed as(2)$${\dot{X}}_{r}\sin\varphi -{\dot{Y}}_{r}\cos\varphi =0$$

By combining the two equations above, the following can be obtained:(3)$$\left\{\begin{array}{l} {\dot{X}}_{r}=v_{r}\cos\varphi \\ {\dot{Y}}_{r}=v_{r}\sin\varphi \end{array}\right.$$

According to the geometric relationship between the front wheels and rear wheels, the following can be obtained:(4)$$\left\{\begin{array}{l} X_{f}=X_{r}+l\cos\varphi \\ Y_{f}=Y_{r}+l\sin\varphi \end{array}\right.$$

At the center M of the vehicle’s rear axle, the yaw rate can be expressed as(5)$$\omega =\frac{v_{r}}{l}\tan\delta_{f}$$

In the formula, ω represents the vehicle yaw rate; meanwhile, the steering radius *R* and the front-wheel steering angle *δ_f_* can be derived from the yaw rate *ω* and the vehicle’s rear-axle speed *v_r_*:(6)$$\left\{\begin{array}{l} R=v_{r}/\omega \\ \delta_{f}=\arctan\left(l/R\right) \end{array}\right.$$

By synthesizing the above equations, the kinematic model of a single vehicle can be obtained as follows:(7)$$\left[\begin{array}{l} {\dot{X}}_{r}\\ {\dot{Y}}_{r}\\ \dot{\varphi } \end{array}\right]=\left[\begin{array}{l} \cos\varphi \\ \sin\varphi \\ \left(\tan\delta_{f}\right)/l \end{array}\right]v_{r}$$

This model can be further expressed in a more general form:(8)$${\dot{\xi }}_{\mathrm{kin}}=f_{\mathrm{kin}}\left(\xi_{\mathrm{kin}},u_{\mathrm{kin}}\right)$$

In this system, the state variables are selected as the longitudinal position, lateral position, and heading angle of the host vehicle in the inertial coordinate system. $\xi_{\mathrm{kin}}={\left[\begin{array}{ccc} X_{r} & Y_{r} & \varphi \end{array}\right]}^{T}$, The control variables are selected as the vehicle speed and the front-wheel steering angle of the vehicle. $u_{\mathrm{kin}}={\left[\begin{array}{cc} v_{r} & \delta_{f} \end{array}\right]}^{T}$; In the formula, $f_{\mathrm{kin}}\left(\begin{array}{cc} \cdot , & \cdot \end{array}\right)$ is the state transition function of the system.

### 2.2. Kinematic Model of Surrounding Vehicles

Considering that there are surrounding vehicles during the vehicle’s driving process, and the motion state of each vehicle is also dynamically updated. The kinematic model for surrounding vehicles is established using the same method as described above:(9)$$\left[\begin{array}{l} {\dot{X}}_{s_{i}}\\ {\dot{Y}}_{s_{i}}\\ {\dot{\varphi }}_{s_{i}} \end{array}\right]=\left[\begin{array}{l} \cos\varphi_{s_{i}}\\ \sin\varphi_{s_{i}}\\ \left(\tan\delta_{{f\_s}_{i}}\right)/l \end{array}\right]v_{s_{i}}$$

In the formula, the subscript $s_{i}$ represents surrounding vehicles, and when i takes values p,lf,lr,rf,rr, they represent the preceding vehicle in the current lane, the preceding vehicle in the left lane, the following vehicle in the left lane, the preceding vehicle in the right lane, and the following vehicle in the right lane, respectively.

Similarly, it can be expressed in a more general form:(10)$${\dot{\xi }}_{{\mathrm{kin}\_s}_{i}}=f_{{\mathrm{kin}\_s}_{i}}\left(\xi_{{\mathrm{kin}\_s}_{i}},u_{{\mathrm{kin}\_s}_{i}}\right)$$

In this system, the state variables are selected as the longitudinal-lateral positions and heading angles of other surrounding vehicles in the inertial coordinate system, $\xi_{{\mathrm{kin}\_s}_{i}}={\left[\begin{array}{ccc} X_{s_{i}} & Y_{s_{i}} & \varphi_{{\_s}_{i}} \end{array}\right]}^{T}$, and the control variables are selected as the vehicle speeds and front-wheel steering angles of these vehicles $u_{{\mathrm{kin}\_s}_{i}}={\left[\begin{array}{cc} v_{s_{i}} & \delta_{{f\_s}_{i}} \end{array}\right]}^{T}$.

## 3. Implementation of Dynamic Trajectory Planning Based on Model Predictive Control

The implementation process of the model predictive theory mainly consists of three key steps [26]: predictive model, rolling optimization, and feedback correction (Figure 3). The predictive model is the foundation of model predictive control, and its main function is to predict the future output of the system based on the historical information of the object and future inputs. Rolling optimization refers to the process where model predictive control determines the control action through the optimization of a certain performance index. However, this optimization is not carried out offline once, but repeatedly online, which is the fundamental difference between model predictive control and traditional optimal control. Feedback correction is intended to prevent the deviation of the control from the ideal state caused by model mismatch or environmental interference. At a new sampling moment, the actual output of the object is first detected, and this real-time information is used to correct the prediction result based on the model, followed by the execution of a new round of optimization.

### 3.1. Vehicle Model Construction—Predictive Model

In the prediction horizon, both the system’s future state variables and output variables can be calculated using the system’s current state variables and the control increments within the control horizon. This corresponds to the “prediction” function in the model predictive algorithm.$$\begin{array}{l} X\left(k+1\right)=X\left(k\right)+\Delta h\left(v\left(k\right)\cos\varphi \left(k\right)\right)\\ Y\left(k+1\right)=Y\left(k\right)+\Delta h\left(v\left(k\right)\sin\varphi \left(k\right)\right)\\ \varphi \left(k+1\right)=\varphi \left(k\right)+\Delta h\left(v\left(k\right)\frac{\tan\delta_{f}\left(k\right)}{L}\right) \end{array}$$

In the formula, k represents a certain moment during the lane-changing process. $X\left(k\right)$, $Y\left(k\right)$, $\varphi \left(k\right)$, $v\left(k\right)$, $\delta_{f}\left(k\right)$ are the values of the vehicle’s driving state variables and control variables at moment k, respectively, Δh is the simulation time step.

Expressed in matrix form:$$x\left(k+1\right)=A\left(k\right)x\left(k\right)+B\left(k\right)u\left(k\right)$$

In the formula,$$A\left(k\right)=\left[\begin{array}{ccc} 1 & 0 & 0\\ 0 & 1 & 0\\ 0 & 0 & 1 \end{array}\right],B\left(k\right)=\left[\begin{array}{cc} \Delta h\cos\varphi \left(k\right) & 0\\ \Delta h\sin\varphi \left(k\right) & 0\\ \Delta h\left(\frac{\tan\delta_{f}\left(k\right)}{L}\right) & 0 \end{array}\right],u\left(k\right)=\left[\begin{array}{c} v\left(k\right)\\ \delta_{f}\left(k\right) \end{array}\right].$$

### 3.2. Establishment of Objective Function with Constraints—Rolling Optimization

At each moment in MPC, optimization must be performed to obtain the optimal control quantity for the next moment (Figure 4).

Objective Function (Performance Function)

Minimize the lateral acceleration of the vehicle during the lane-changing process to ensure riding comfort: $J_{c}=a_{y}$. In the formula, $a_{y}$ represents the lateral acceleration value during the lane-changing process.Enable the lane change to be completed as quickly as possible, with the shortest possible lane-changing time: $J_{t}=y_{d}-y_{e}$. In the formula, $y_{d}$ and $y_{e}$ represent the desired lateral displacement and the actual lateral displacement during the lane-changing process, respectively.

Convert each index into a quadratic form to obtain the total objective function (performance index function):$$\begin{array}{cc} \min J & =J_{c}+J_{t}\\ & =\omega_{1}\left(\int_{t_{0}}^{t_{f}}{\left(y_{d}-y_{e}\right)}^{2}dt\right)+\omega_{2}\left(a_{y}^{2}\right) \end{array}$$

In the formula, $\omega_{1}$ and $\omega_{2}$ are the weight coefficients of each index in the objective function; $y_{d}$ is the desired lateral displacement during the lane-changing process, which is set as the lane width in this paper, taking 3.5 m; $y_{e}$ is the actual lateral displacement during the lane-changing process; $y_{e}=y_{0}+\int_{t_{0}}^{t_{f}}v\sin\varphi dt$ and $y_{0}$ are the vehicle lateral position information at the start moment of lane change; $a_{y}$ is the vehicle lateral acceleration value during the lane-changing process, which can be expressed as $a_{y}=\frac{v^{2}}{L\left(1+Kv^{2}\right)}\delta_{f}$ according to the stable yaw rate gain relationship of the two-degree-of-freedom vehicle and the motion relationship of the vehicle during instantaneous curve driving.

From this, it can be concluded that the expression of the objective function is only related to the vehicle driving state $x=\left[\varphi \right]$ and the control quantity $u=\left[v,\delta_{f}\right]$.

In the above relational expression, v is the vehicle’s driving speed; φ is the vehicle’s heading angle; $\delta_{f}$ is the vehicle’s front-wheel steering angle; K is the vehicle’s stability factor, $K=\frac{m}{L^{2}}\left(\frac{a}{C_{r}}-\frac{b}{C_{f}}\right)$, with the unit of $s^{2}{/m}^{2}$; where $C_{f}$ and $C_{r}$ are the cornering stiffness of the front and rear tires, respectively; and L is the vehicle’s wheelbase.

2.Constraints

Ensure that the relative position between the Ego vehicle and surrounding vehicles remains appropriate during the lane-changing process—position constraint:


$$J_{d}=\sum_{i=1}^{s_{i}}\sqrt{{\left(x_{e}-x_{i}\right)}^{2}+{\left(y_{e}-y_{i}\right)}^{2}}\geq S_{safe}$$


In the formula, i and n represent the ID number of surrounding vehicles and the number of surrounding vehicles, respectively; $\left(x_{i},y_{i}\right)$ is the position coordinate information of the i-th surrounding vehicle, which can be obtained in real time through on-board sensors; $J_{d}$ represents the real-time distance between the Ego vehicle and surrounding vehicles; $S_{safe}$ represents the minimum safe distance between the Ego vehicle and surrounding vehicles during driving.

Satisfy the equality constraint equation for changes in vehicle driving states—equation constraint:


$$x_{t+1}=A_{t}x_{t}+B_{t}u_{t}$$


The position coordinate information of the future predicted output must fall within the planned feasible region—output constraint:


$$X_{\min}\leq X\leq X_{\max},Y_{\min}\leq Y\leq Y_{\max}$$


The vehicle speed and front-wheel steering angle must be controlled within the executable range of the vehicle—control constraint:


$$v_{\min}\leq v\leq v_{\max},\Delta v_{\min}\leq \Delta v\leq \Delta v_{\max}$$



$$\delta_{f\_\min}\leq \delta_{f}\leq \delta_{f\_max},\Delta \delta_{f\_\min}\leq \Delta \delta_{f}\leq \Delta \delta_{f\_max}$$


Through the above analysis, the problem is converted into an optimization solution problem for the objective function with constraints:
(11)$$\begin{array}{c} \underset{u_{t}}{\min}J=\omega_{1}\left(\int_{t_{0}}^{t_{f}}{\left(y_{d}-y_{e}\right)}^{2}dt\right)+\omega_{2}\left(a_{y}^{2}\right)\\ s.t.\\ \sum_{i=1}^{s_{i}}\sqrt{{\left(x_{e}-x_{i}\right)}^{2}+{\left(y_{e}-y_{i}\right)}^{2}}\geq S_{safe}\\ x_{t+k+1}=Ax_{t+k}+Bu_{t+k}\\ x_{t+k}\in X,\;k=0,\ldots ,N\\ u_{t+k}\in U,\;k=0,\ldots ,N-1 \end{array}$$

By solving this optimization objective with constraints, the control sequence $u^{*}\left(k\right)={\left[\begin{array}{cc} v^{*}\left(k\right) & \delta_{f}^{*}\left(k\right) \end{array}\right]}^{T}$ for a future period can be obtained. However, since the system model changes in real time, it cannot be guaranteed that a feasible solution for this optimization objective can be obtained at every moment. Therefore, the optimization objective needs to be processed accordingly.

A common and proven effective method is to add a slack variable to the optimization objective:(12)$$\underset{u_{t}}{\min}J=\omega_{1}\left(\int_{t_{0}}^{t_{f}}{\left(y_{d}-y_{e}\right)}^{2}dt\right)+\omega_{2}\left(a_{y}^{2}\right)+\rho \varepsilon^{2}$$

In the formula, ρ is the weight coefficient, and ε is the slack variable.

3.Coordinate Conversion of Control Quantities

To achieve the optimal solution for the established multi-objective function and constraint conditions, and in consideration of the vehicle’s driving characteristics, the area that the vehicle can reach at the next moment—predictable based on the vehicle’s speed and heading angle information—can be represented by a sector. For standardized representation and to facilitate the direct solution of control quantities, a conversion method is proposed to convert the 2D coordinate system into a polar coordinate form. This method converts the speed and heading angle (which are the corresponding control quantities) into the radius and rotation angle forms of polar coordinates (Figure 5).

Correspondingly, the vehicle speed v is represented as the radius R in polar coordinates, and the front-wheel steering angle $\delta_{f}$ is represented as the rotation angle θ in polar coordinates. The coordinate relational expressions corresponding to the vehicle’s driving process are as follows:(13)$$\left\{\begin{array}{l} x_{e}=R\cos\left(\theta \right)\\ y_{e}=R\sin\left(\theta \right) \end{array}\right.$$

In this way, the optimal solution of the multi-objective function with constraints is converted into the solution of *R* and *θ* in the polar coordinate system.

### 3.3. Return of the Optimal Control Sequence—Feedback Adjustment

After completing the optimal solution of the above formula in each control cycle, a series of control input increments within the control horizon are obtained.$$\Delta U_{kin,t}^{*}={\left[\Delta u_{kin,t}^{*},\Delta u_{kin,t+1}^{*},\ldots ,\Delta u_{kin,t+N_{c}-1}^{*}\right]}^{T}$$

According to the basic principle of model predictive control, the first element in the control sequence is used as the actual control input increment acting on the system, that is, $u_{kin}\left(t\right)=u_{kin}\left(t-1\right)+\Delta u_{kin,t}^{*}$.

The system executes this control quantity until the next moment. At the new moment, the system re-predicts the output of the next time domain based on the state information, and obtains a new sequence of control increments through the optimization process. This cycle repeats until the system completes the entire control process.

### 3.4. Curve Fitting of Optimal Discrete Points

The discrete points corresponding to the optimal solution obtained from the rolling optimization process are subjected to curve fitting to plan a smooth and feasible trajectory line. To fully ensure the smoothness of the curve and that the curve curvature at the start and end points can meet the consistency of the vehicle’s forward direction, a quintic polynomial is used for least squares fitting.

The least squares fitting polynomial (also known as the method of least squares) is a mathematical optimization technique. It finds the best function fit for the data by minimizing the sum of the squares of the errors.

Given the function value $y_{1},\ldots ,y_{n}$ of function $y=f\left(x\right)$ at point $x_{1},\ldots ,x_{n}$, find a polynomial $p\left(x\right)=a_{0}+a_{1}x+\ldots +a_{m}x^{m}\in \prod_{m}\left(m+1<n\right)$ such that $\min\left(\sum_{i=1}^{n}{\left(p\left(x_{i}\right)-y_{i}\right)}^{2}\right)$

Let $S\left(a_{0},a_{1},\ldots ,a_{m}\right)=\sum_{i=1}^{n}{\left(a_{0}+a_{1}x_{i}+\ldots +a_{m}x_{i}^{m}-y_{i}\right)}^{2}$

Solving $\min S\left(a_{0},a_{1},\ldots ,a_{m}\right)$ is thus converted into solving$$\frac{\partial S\left(a_{0},a_{1},\ldots ,a_{m}\right)}{\partial a_{j}}=\sum_{i=1}^{n}2\left(a_{0}+a_{1}x_{i}+\ldots +a_{m}x_{i}^{m}-y_{i}\right)x_{i}^{j}=0$$

After rearranging the equation, we obtain$$\left\{\begin{array}{l} na_{0}+\left(\sum_{i=1}^{n}x_{i}\right)a_{1}+\ldots +\left(\sum_{i=1}^{n}x_{i}^{m}\right)a_{m}=\sum_{i=1}^{n}y_{i}\\ \left(\sum_{i=1}^{n}x_{i}\right)a_{0}+\left(\sum_{i=1}^{n}x_{i}^{2}\right)a_{1}+\ldots +\left(\sum_{i=1}^{n}x_{i}^{m+1}\right)a_{m}=\sum_{i=1}^{n}x_{i}y_{i}\\ \ldots \ldots \\ \left(\sum_{i=1}^{n}x_{i}^{m}\right)a_{0}+\left(\sum_{i=1}^{n}x_{i}^{m+1}\right)a_{1}+\ldots +\left(\sum_{i=1}^{n}x_{i}^{2m}\right)a_{m}=\sum_{i=1}^{n}x_{i}^{m}y_{i} \end{array}\right.$$

Solving this system of normal equations yields the value of $a_{0},a_{1},\ldots ,a_{m}$.

Thus, the expression of the least squares fitting polynomial for $f\left(x\right)$ is obtained.$$p\left(x\right)=a_{0}+a_{1}x+\ldots +a_{m}x^{m}.$$

According to the principle of model predictive control, the prediction horizon Np is set to 5 to predict the optimal points at the next 5 moments. Together with the initial position point $\left(x_{0},y_{0}\right)$ of the vehicle at the initial moment, there are 6 points in total. The curve fitting toolbox in MATLAB(R2022a) is used to perform quintic polynomial curve fitting, so as to obtain the real-time optimal trajectory. To facilitate the direct control of the vehicle, this study converts the planned optimal trajectory into a steering wheel angle signal based on the preview-following driver model [27].

## 4. Simulation and Analysis

A co-simulation is conducted using MATLAB/Simulink and Prescan (Figure 6). A driving trajectory generator is built in MATLAB, while a driving scenario is constructed in Prescan. The planned trajectory is converted into a front-wheel steering wheel angle signal via the preview-following driver module, which is then input to the controlled vehicle. This process is used to perform simulation verification of the proposed method.

### 4.1. The Surrounding Vehicles Maintain a Constant Speed

This is the simulation of the lane-change trajectory realized by the proposed model predictive control method when surrounding vehicles maintain a constant speed. The detailed process of the simulation working condition is shown in the Figure 7. In the simulation, the speed of the host vehicle is set to 12 m/s; the speed of the preceding vehicle is 10 m/s, and it keeps moving straight; the speed of the vehicle behind on the left is 8 m/s, and it also keeps moving straight.

The Figure 7 shows the lane-change scenario in a dynamic environment, which occurs when the preceding vehicle’s speed is detected to be lower than the host vehicle’s speed and the lane-change conditions are met. Figure 7a illustrates the initial state just before lane change begins, while Figure 7b shows the situation where the steering wheel angle signal is 0 at this moment. Figure 7c depicts the process of the host vehicle merging from the original lane into the target lane during the lane change, with the corresponding steering wheel turning status shown in Figure 7d. Figure 7e presents the scenario where the host vehicle travels along the target lane after entering it, and the corresponding steering wheel angle also returns to 0. Figure 7f shows the driver’s forward-facing view after the host vehicle has fully completed the lane-change maneuver

As can be seen from Figure 8, Figure 9, Figure 10 and Figure 11, when the host vehicle approaches the preceding vehicle and the preceding vehicle’s speed (10 m/s) is lower than that of the host vehicle (12 m/s), the host vehicle will generate an intention to change lanes. At this point, if a safe lane-change distance from the left-behind vehicle is detected, the host vehicle will start the lane change.

The entire lane-change process lasts 5.2 s, covering a distance of 47 m, with a maximum steering wheel angle of 41 degrees during the process. At the start of the lane change, the initial relative distances between the host vehicle and the preceding vehicle, and between the host vehicle and the left-behind vehicle are 18.3 m and 12.6 m, respectively. Since the host vehicle’s speed is higher than that of the left-behind vehicle, the relative distance between them keeps increasing until it reaches 20 m (the measurement range of the sensor), and a relative distance greater than 20 m is maintained throughout the entire lane-change process.

Meanwhile, as the host vehicle’s speed is higher than that of the preceding vehicle, the relative distance between them keeps decreasing until the lane change is completed. After entering the target lane, there is no following vehicle ahead, so the corresponding relative distance value becomes 0.

Combined with the above simulation results, it can be concluded that the designed lane-change trajectory planner enables a smooth and safe lane-change process.

### 4.2. Sudden Deceleration of the Preceding Vehicle—Constant Speed of the Left-Behind Vehicle

This is the simulation and analysis of the lane-change trajectory realized by the proposed model predictive control method under the dynamic condition of surrounding vehicles, where the preceding vehicle decelerates suddenly while the left-behind vehicle maintains a constant speed. The detailed process of the simulation condition is shown in the Figure 12.

In the simulation, the host vehicle’s speed is set to 12 m/s. The preceding vehicle has an initial speed of 10 m/s; after running for 1.5 s, it suddenly applies a braking acceleration of 2.5 m/s^2^ and continues to move straight. The left-behind vehicle maintains a constant speed of 8 m/s and travels straight.

In this simulation scenario, the host vehicle intends to perform a lane change. However, during the initial phase of the lane change, it suddenly detects a sudden speed reduction of the preceding vehicle. Nevertheless, the lane-change conditions are still satisfied, so the lane change proceeds in this dynamic environment.

Figure 12a shows the initial state just before the lane change starts, and Figure 12b presents the situation where the steering wheel angle signal is 0 at this moment. In Figure 12c, during the lane-change process, the host vehicle detects a sudden deceleration of the preceding vehicle. To respond to this situation, the planned lane-change trajectory enables the host vehicle to complete the lane change earlier and more quickly to ensure driving safety, with the corresponding steering wheel turning status shown in Figure 12d. Figure 12e depicts the scenario where the host vehicle travels along the target lane after entering it, and the corresponding steering wheel angle also returns to 0. Figure 12f shows the driver’s-view perspective after the lane change has been completed.

As can be seen from Figure 13, Figure 14, Figure 15 and Figure 16, when the host vehicle approaches the preceding vehicle and the preceding vehicle’s speed (10 m/s) is lower than that of the host vehicle (12 m/s), the host vehicle generates an intention to change lanes. At this point, if a safe lane-change distance from the left-behind vehicle is detected simultaneously, the host vehicle starts the lane change. When the host vehicle is traveling at 12 m/s, it initiates the lane change after 1.2 s of driving. 0.3 s after the start of the maneuver, a sudden speed reduction of the preceding vehicle is detected. To ensure safety, the host vehicle applies a larger steering wheel angle to achieve a rapid lane change. Under this condition, the maximum steering wheel angle is 61 degrees, the entire lane-change process lasts 4.6 s, and the lane-change distance is 42 m.

The host vehicle has an initial speed of 12 m/s; during the lane change, its speed decreases slightly due to the detected speed reduction in the preceding vehicle. The preceding vehicle first travels at 10 m/s, then suddenly applies a braking acceleration of 2.5 m/s^2^ after 1.5 s, causing its speed to drop rapidly. After 3.3 s of driving, the preceding vehicle’s speed decreases to 5 m/s and remains constant as it continues straight. The left-behind vehicle maintains a constant speed of 8 m/s while traveling straight. Throughout the process, the host vehicle travels 87 m longitudinally, with a lateral displacement of 3.5 m (from the original lane to the target lane); the preceding vehicle travels 46 m, and the left-behind vehicle travels 56 m.

At the start of the lane change, the initial relative distances between the host vehicle and the preceding vehicle, and between the host vehicle and the left-behind vehicle are 18.1 m and 12.4 m, respectively. Since the host vehicle’s speed is higher than that of the left-behind vehicle, the relative distance between them keeps increasing until it reaches 20 m (the measurement range of the sensor), and a relative distance greater than 20 m is maintained throughout the entire lane-change process. Meanwhile, as the host vehicle’s speed is higher than that of the preceding vehicle, the relative distance between them keeps decreasing. When the preceding vehicle starts to decelerate, the relative distance between the host vehicle and the preceding vehicle drops rapidly; by 3.2 s, this relative distance becomes 0, indicating that the host vehicle has completed the lane change.

Combined with the above simulation results, it can be observed that in response to the lane-change scenario where the preceding vehicle decelerates suddenly, the host vehicle completes the lane change in advance and applies a larger steering wheel angle signal to achieve a faster lane-change maneuver. This also verifies that the designed lane-change trajectory planner can well adapt to the special condition of sudden speed reduction of the preceding vehicle.

### 4.3. Sudden Acceleration of the Left-Behind Vehicle—Constant Speed of the Preceding Vehicle

When surrounding vehicles are in a dynamic condition and the host vehicle detects that the left-behind vehicle suddenly accelerates after initiating a lane change, the proposed model predictive control method is subjected to simulation analysis of the lane-change trajectory under this condition. The detailed process of the simulation condition is shown in the Figure 17. In the simulation, the initial speed of the host vehicle is set to 12 m/s; the preceding vehicle travels at a constant speed of 10 m/s in a straight line; the left-behind vehicle initially travels at 8 m/s, and after running for 1.5 s, it suddenly receives an acceleration of 2.5 m/s^2^, accelerating to 14 m/s and then maintaining this speed for straight-line travel.

In this simulation scenario, the host vehicle is performing a lane-change maneuver. However, after the lane change starts, it suddenly detects that the speed of the left-behind vehicle increases abruptly and the relative distance between the host vehicle and the left-behind vehicle decreases suddenly—this is the lane-change scenario in a dynamic environment under such conditions.

Figure 17a shows the state where the host vehicle has a lane-change demand and is about to start the lane change; Figure 17b presents the situation where the steering wheel angle signal is 0 at this moment. In Figure 17c, during the lane-change process, the host vehicle detects that the left-behind vehicle accelerates suddenly and the relative distance between them decreases abruptly. To respond to this situation, the proposed lane-change trajectory planner enables the host vehicle to increase its speed accordingly during the lane change and complete the lane change more quickly, thus ensuring driving safety; the corresponding steering wheel turning status is shown in Figure 17d. Figure 17e depicts the scenario where after the host vehicle enters the target lane and travels along it, the corresponding steering wheel angle also returns to 0. Figure 17f shows the driver’s forward-looking view after the lane-change maneuver has been completed.

As can be seen from Figure 18, Figure 19, Figure 20 and Figure 21, when the host vehicle approaches the preceding vehicle and the preceding vehicle’s speed (10 m/s) is lower than that of the host vehicle (12 m/s), the host vehicle will generate an intention to change lanes. At this point, if a safe lane-change distance from the left-behind vehicle is detected simultaneously, the host vehicle will start the lane change.

The host vehicle initially travels straight along the lane at 12 m/s for 1.4 s; when the relative distance to the preceding vehicle is 15 m, it begins the lane-change maneuver. 2.1 s after the start of the lane change, the host vehicle detects a sudden decrease in the relative distance to the left-behind vehicle. To ensure safety, the host vehicle increases its speed accordingly to complete the lane change quickly and safely. Under this condition, the maximum steering wheel angle is 43 degrees, the entire lane-change process lasts 4.7 s, and the lane-change distance is 48 m.

From the vehicle speed and position diagrams, it can be observed that the preceding vehicle travels at a constant speed of 10 m/s in a straight line throughout the process. The left-behind vehicle has an initial speed of 8 m/s; after traveling for 1.5 s, it suddenly accelerates with an acceleration of 2.5 m/s^2^, reaching 14 m/s after 2.4 s, and then maintains this speed for straight-line travel—during this period, the relative distance between the left-behind vehicle and the host vehicle keeps decreasing. The host vehicle initially travels at 12 m/s; as the lane change starts and the left-behind vehicle accelerates, when the host vehicle detects that the relative distance to the left-behind vehicle is less than 20 m at 3.8 s, it begins to accelerate until the lane change is completed and the simulation ends. Throughout the process, the host vehicle travels 95 m longitudinally with a lateral displacement of 3.5 m (from the original lane to the target lane); the preceding vehicle travels 75 m, and the left-behind vehicle travels 86 m.

At the start of the lane-change process, the initial relative distances between the host vehicle and the preceding vehicle, and between the host vehicle and the left-behind vehicle are 18.1 m and 12.4 m, respectively. Since the host vehicle’s speed is higher than that of the preceding vehicle, the relative distance between them keeps decreasing until it drops to 15 m—at this point, a lane-change demand is generated, and the host vehicle starts the lane change. Initially, the left-behind vehicle’s speed is lower than that of the host vehicle, so the relative distance between them keeps increasing. However, the left-behind vehicle suddenly accelerates after traveling for 1.5 s; by 3.8 s, the host vehicle detects that the relative distance to the left-behind vehicle is less than 20 m (note: corrected from “20 m/s” to “20 m” for unit consistency), so the host vehicle starts to accelerate. Correspondingly, the relative distance first decreases slightly and then keeps increasing until it returns to the safe range of more than 20 m at 6.8 s.

From the above simulation results, it can be concluded that in response to the lane-change scenario where the left-behind vehicle accelerates suddenly, when the host vehicle detects that the relative distance to the left-behind vehicle is less than the safe distance, the designed trajectory planner controls the host vehicle to accelerate to re-enter the safe range. This verifies that the designed lane-change trajectory planner can well adapt to the special condition of sudden acceleration of the left-behind vehicle.

Based on the comprehensive results of the aforementioned simulation experiments, it can be indicated that the innovative real-time optimal iteration improvement of the MPC method proposed in this paper enables the real-time search for the optimal position of the vehicle at the next moment. Consequently, an ideal and feasible lane-changing trajectory for dynamic driving scenarios can be planned, achieving adaptive performance for the entire range of dynamic driving scenarios.

## 5. Conclusions

This paper conducts research on lane-change trajectory planning in dynamic scenarios. First, it establishes the kinematic models of the host vehicle and surrounding vehicles, which are then converted into state equations to serve as the prediction model. Next, a multi-objective function including three indicators—characterizing lane-change rapidity, safety, and comfort—is constructed, and constraint conditions such as output, input, and road boundaries are established to further complete the rolling optimization link of model predictive control (MPC). For the convenience of solving, the control variables are converted from a 2D coordinate system to a polar coordinate system. The optimal discrete points of the control variables obtained through optimization are fitted with a quintic polynomial curve to generate a smooth and feasible lane-change trajectory curve. To facilitate the direct control of the vehicle, a preview-following driver model is adopted to convert the planned ideal trajectory into a steering wheel angle signal. Finally, simulation verification of the proposed method is carried out under three working conditions: surrounding vehicles traveling at a constant speed, sudden deceleration of the preceding vehicle during lane change, and sudden acceleration of the left-behind vehicle during lane change. The simulation results show that the dynamic trajectory planning method proposed in this paper has good adaptability.

This paper comprehensively considers complex lane-change scenarios involving a preceding vehicle and a left-behind vehicle, and proposes a dynamic lane-change trajectory planning method for highway dynamic scenarios based on the model predictive control theory. The proposed method can realize real-time response to the dynamic environment; moreover, polynomial curve fitting of the discrete points obtained after optimization via coordinate conversion can better ensure the vehicle’s track-following performance with higher smoothness, which is more in line with actual driving conditions. This provides a theoretical basis for the realization of lane change in dynamic scenarios for autonomous vehicles.

## Figures and Tables

**Figure 1 sensors-25-07241-f001:**
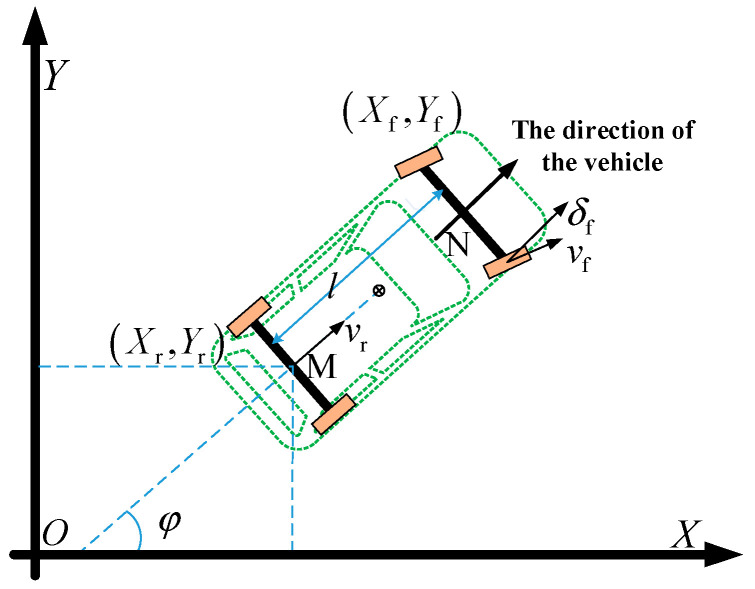
Vehicle motion model.

**Figure 2 sensors-25-07241-f002:**
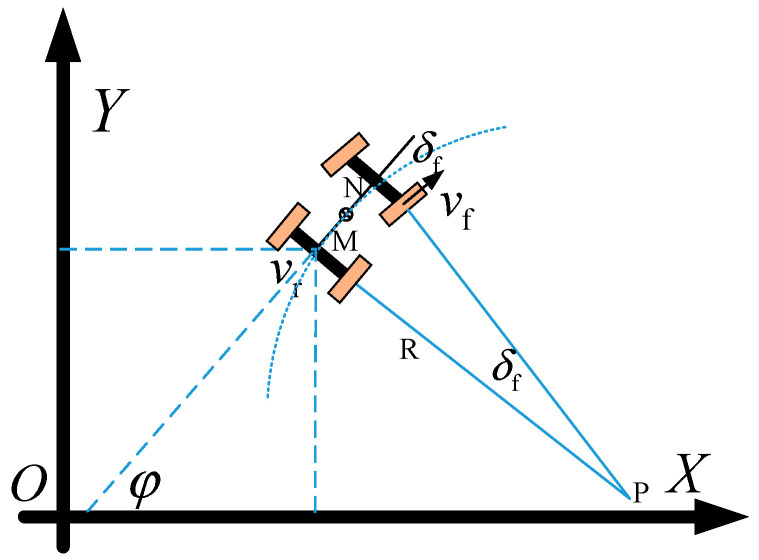
Schematic diagram of vehicle front-wheel steering.

**Figure 3 sensors-25-07241-f003:**
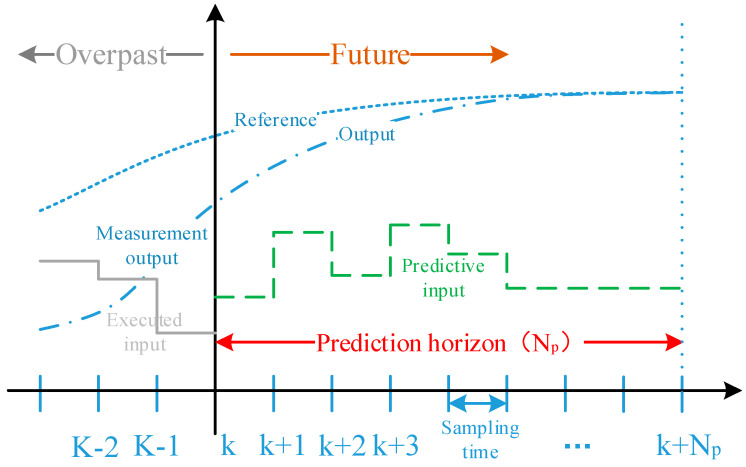
Schematic diagram of model predictive control principle.

**Figure 4 sensors-25-07241-f004:**
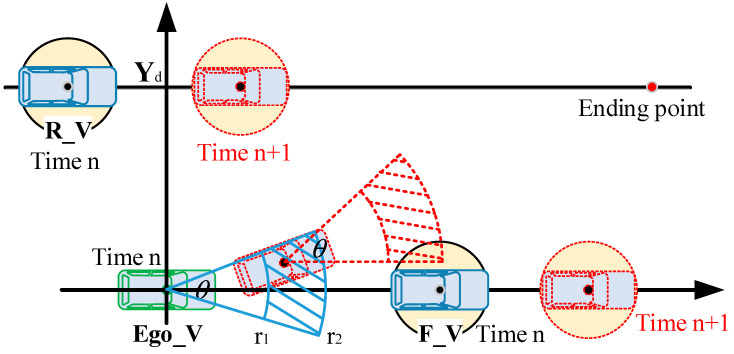
Schematic diagram of lane-change trajectory planning in a dynamic environment.

**Figure 5 sensors-25-07241-f005:**
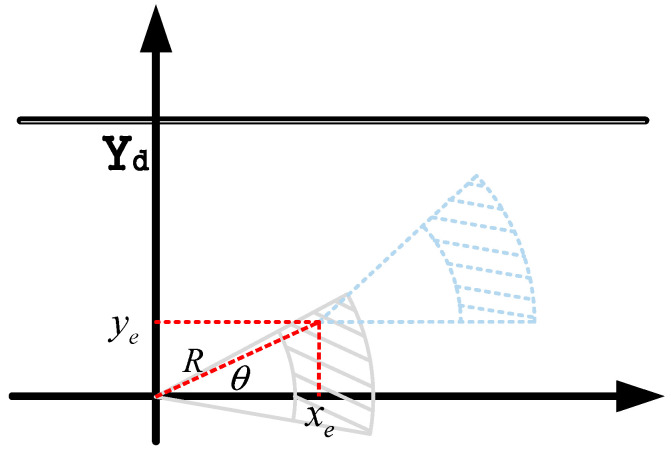
The control quantities are converted into the form of polar coordinates.

**Figure 6 sensors-25-07241-f006:**
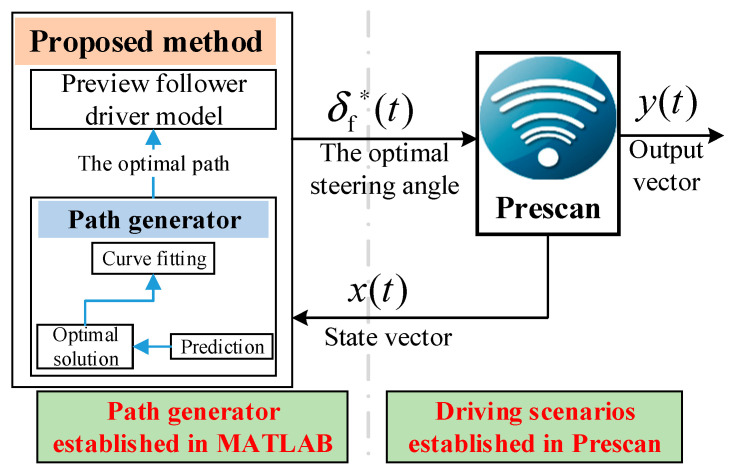
MATLAB/Simulink-Prescan co-simulation structure.

**Figure 7 sensors-25-07241-f007:**
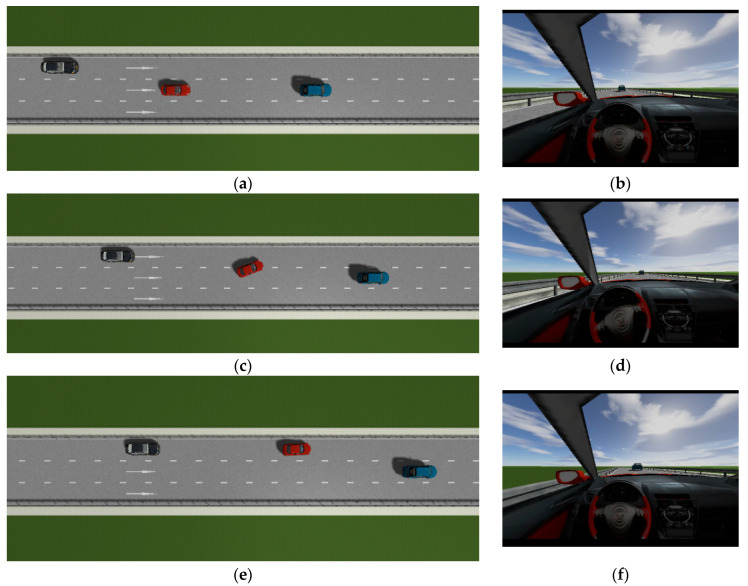
The surrounding vehicles maintain a constant speed simulation scenario.

**Figure 8 sensors-25-07241-f008:**
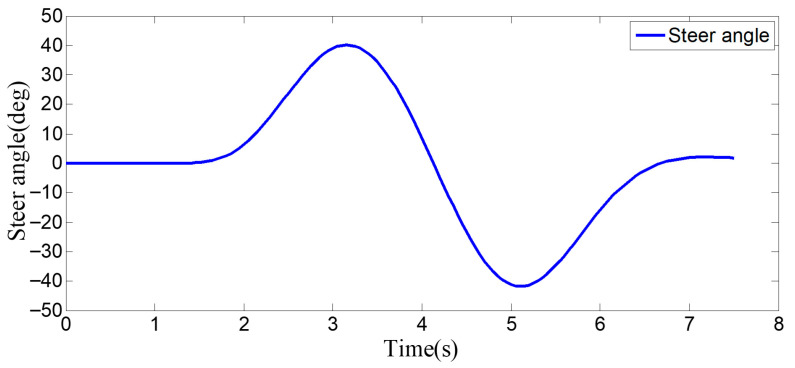
Steering wheel angle signal during the lane-change process when the surrounding vehicles maintain a constant speed.

**Figure 9 sensors-25-07241-f009:**
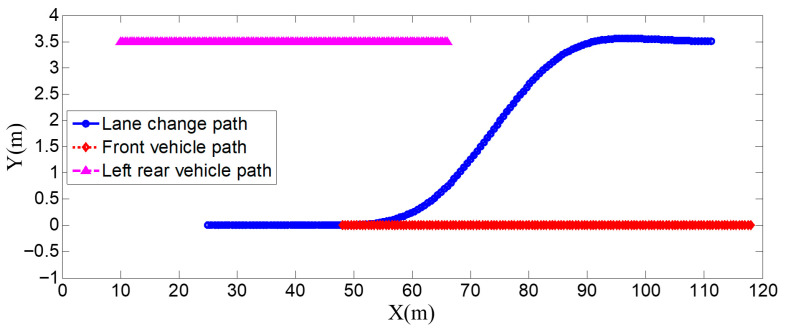
Position curve diagram of the host vehicle and surrounding vehicles during the lane-change process when the surrounding vehicles maintain a constant speed.

**Figure 10 sensors-25-07241-f010:**
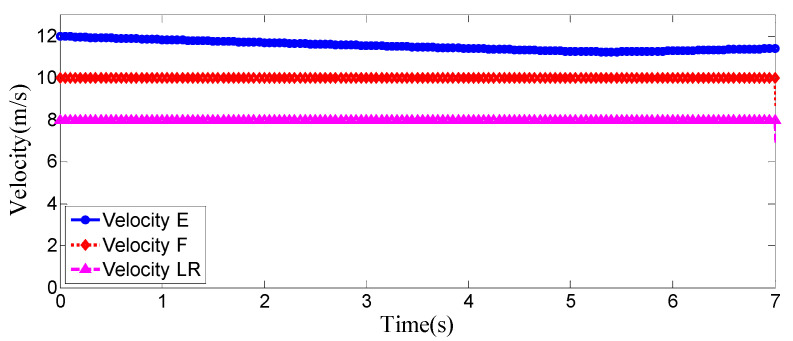
Vehicle speed information of the host vehicle, the preceding vehicle, and the vehicle behind on the left when the surrounding vehicles maintain a constant speed.

**Figure 11 sensors-25-07241-f011:**
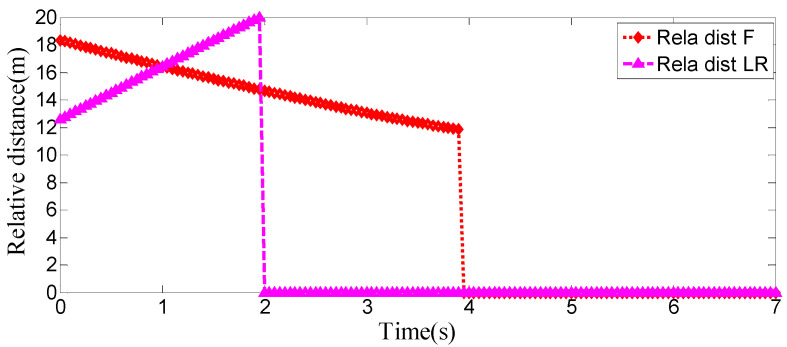
Real-time relative distances: host vehicle vs. preceding vehicle and left-behind vehicle.

**Figure 12 sensors-25-07241-f012:**
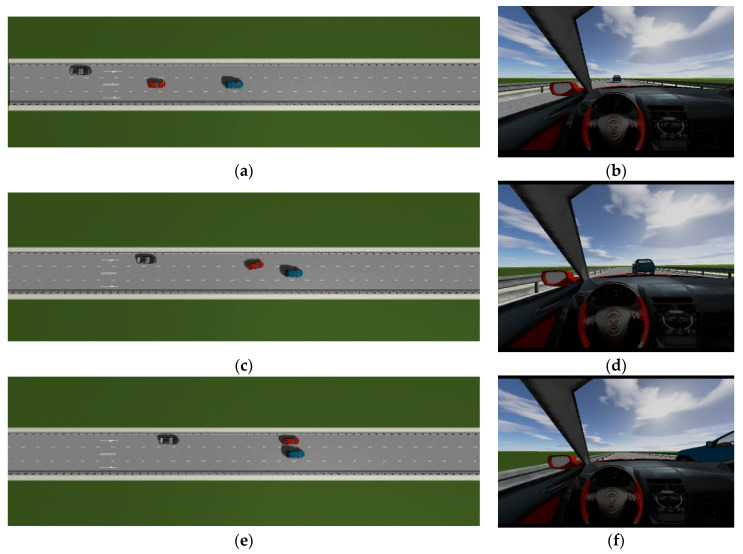
The sudden deceleration of the preceding vehicle simulation scenario.

**Figure 13 sensors-25-07241-f013:**
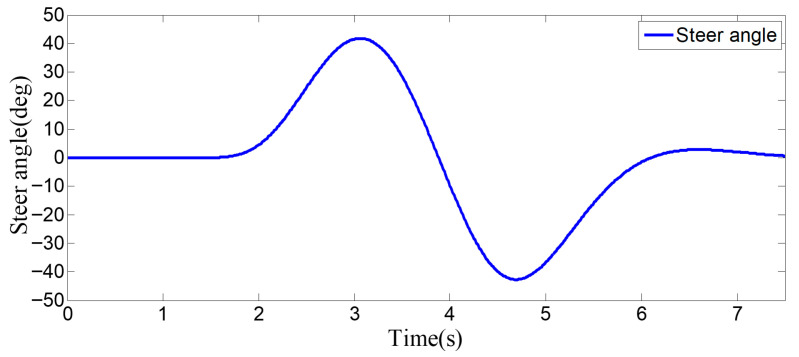
Steering wheel angle signal during the lane-change process when the sudden deceleration of the preceding vehicle.

**Figure 14 sensors-25-07241-f014:**
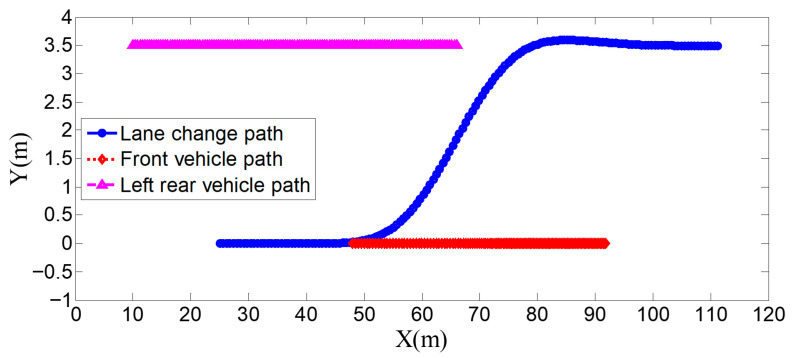
Position curve of the host vehicle and surrounding vehicles during lane change when the sudden deceleration of the preceding vehicle.

**Figure 15 sensors-25-07241-f015:**
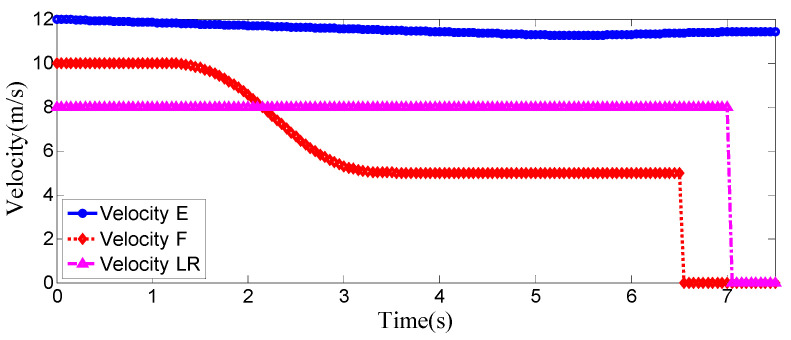
Vehicle speed information of the host vehicle, preceding vehicle and left-behind vehicle when the sudden deceleration of the preceding vehicle.

**Figure 16 sensors-25-07241-f016:**
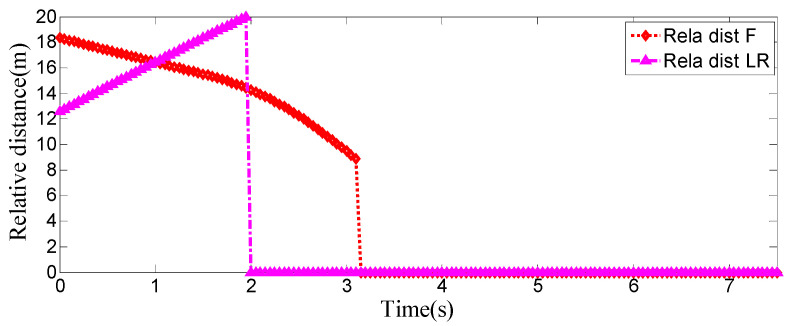
Real-time relative distances: host vehicle vs. preceding and left-behind vehicles.

**Figure 17 sensors-25-07241-f017:**
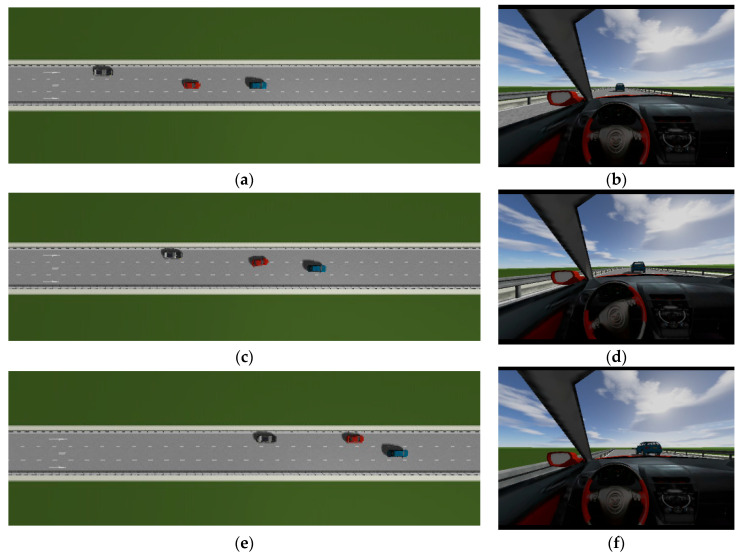
The sudden acceleration of the left-behind vehicle simulation scenario.

**Figure 18 sensors-25-07241-f018:**
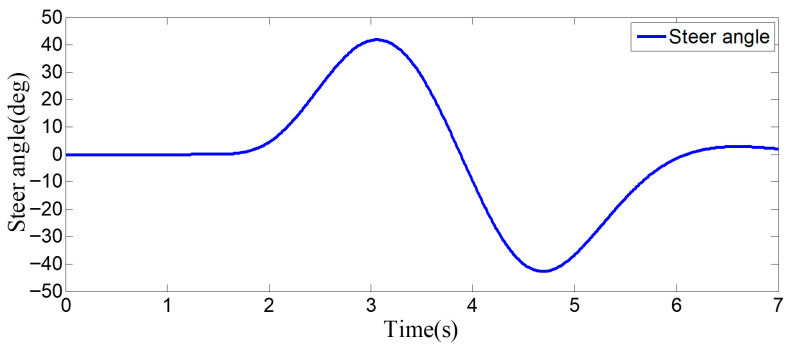
Steering wheel angle signal during the lane-change process when the sudden acceleration of the left-behind vehicle.

**Figure 19 sensors-25-07241-f019:**
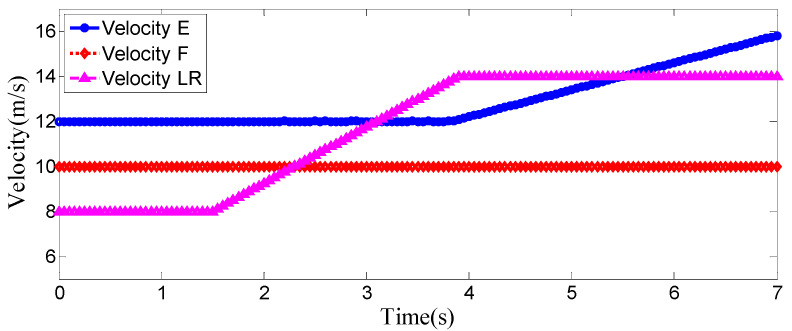
Vehicle speed of the host vehicle, the preceding vehicle, and the left-behind vehicle when the sudden acceleration of the left-behind vehicle.

**Figure 20 sensors-25-07241-f020:**
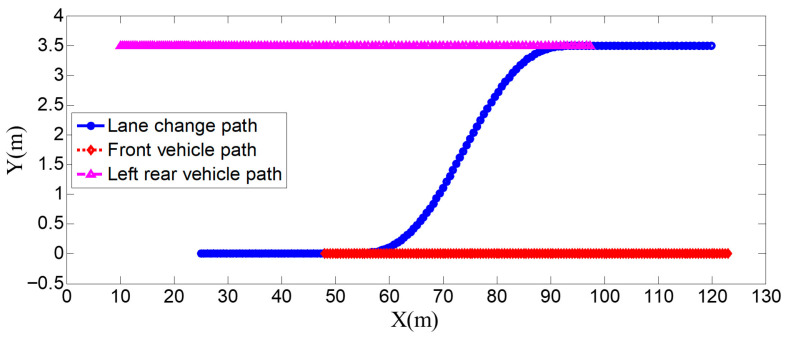
Position curve diagram of the host vehicle and surrounding vehicles during the lane-change process when the sudden acceleration of the left-behind vehicle.

**Figure 21 sensors-25-07241-f021:**
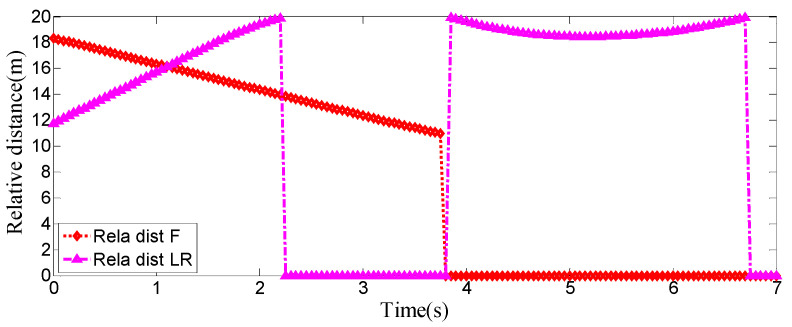
Real-time relative distances between the host vehicle and the preceding vehicle, as well as between the host vehicle and the left-behind vehicle.

## Data Availability

The original contributions presented in this study are included in the article. Simulation scripts and representative datasets can be made available upon reasonable request. Further inquiries can be directed to the corresponding author.

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
