# Peer review of "Research on Dynamic Trajectory Planning Based on Model Predictive Theory for Complex Driving Scenarios"

_sensors, 2025, doi:10.3390/s25237241_

Round 1
Reviewer 1 Report
Comments and Suggestions for Authors
Comments
- The paper addresses lane-change trajectory planning for autonomous vehicles, which is an important and challenging problem. The main focus is investigation how to generate a smooth, feasible, and adaptive lane-change trajectory using model predictive control (MPC) that responds effectively to dynamic traffic conditions.
- The topic is both original and highly relevant to the field of autonomous driving and intelligent vehicle control. Lane-change trajectory planning in dynamic scenarios is a critical challenge for autonomous vehicles, especially when considering real-time interactions with surrounding vehicles. While there has been prior research on trajectory planning, most existing studies focus on simplified or static traffic scenarios and do not fully address the complexities of dynamic, multi-vehicle interactions with sudden acceleration or deceleration events.
- This study addresses a clear gap by proposing a method based on model predictive control (MPC) that integrates multi-objective optimization, real-time adaptability, and trajectory smoothing through coordinate conversion and polynomial fitting. The approach provides a systematic framework for generating safe, comfortable, and feasible lane-change trajectories in highly dynamic traffic conditions, which is not fully covered by current methods. Therefore, the work represents a meaningful contribution to advancing trajectory planning for autonomous vehicles in realistic highway scenarios.
- This paper makes a meaningful contribution to the field of autonomous driving. Compared with previous work, it addresses lane-change trajectory planning in realistic dynamic scenarios, taking into account both preceding and left-behind vehicles and their sudden changes in speed. The method combines model predictive control with multi-objective optimization for safety, comfort, and maneuver speed, and uses coordinate conversion and polynomial fitting to generate smooth, feasible trajectories. The approach is validated under multiple traffic scenarios, showing good adaptability, which makes it more practical and closer to real-world highway conditions than many existing studies.
The methodology is generally sound, but several improvements could strengthen the study:
- Uncertainty and Sensor Noise: Consider incorporating uncertainties in vehicle behavior. This would make the MPC-based trajectory planning more robust to real-world conditions.
- Parameter Sensitivity: Providing an analysis of how sensitive the method is to key parameters (e.g., weights in the objective function, prediction horizon) would help readers understand the method’s stability and reliability.
- The conclusions are consistent with the evidence and arguments presented, and they address the main research question. The paper clearly demonstrates, through simulation studies under multiple dynamic scenarios, that the proposed MPC-based lane-change trajectory planning method can generate smooth, safe, and feasible trajectories. The step-by-step methodology from kinematic modeling, multi-objective optimization, coordinate conversion, to polynomial fitting, is logically connected to the simulation results. The results show that the method adapts to changes in surrounding vehicles’ speeds, maintains safety distances, and achieves smooth lane changes, which directly supports the main question of how to plan lane changes in dynamic highway scenarios in real time. Therefore, the conclusions appropriately reflect the evidence and substantiate the proposed method’s effectiveness.
- The references cited in the manuscript are general appropriate and relevant to the topic of autonomous vehicle trajectory planning and model predictive control (MPC). They cover a range of related areas, including MPC-based lane-change and collision avoidance, artificial potential field methods, sampling-based and optimization-based trajectory planning, reinforcement learning approaches, and kinematic modeling of vehicles. Most of the references are recent (2022–2025), which is good for showing the current state-of-the-art in the field. The inclusion of both journal articles and conference proceedings reflects the applied and experimental nature of trajectory planning research.
- One minor point for improvement is that a few references are repeated (e.g., Yang et al., 2024, on MPC with adaptive APF appears twice), and the authors might consider removing duplicates. Overall, the references adequately support the methodology, discussion, and conclusions, and they demonstrate that the authors are aware of relevant prior work and research gaps. All figure are clear, well-labeled, and effectively support the methodology and results. No issues were observed, and they adequately illustrate the key points of the study.
Comments on the Quality of English Language
The English in the manuscript is generally clear. The ideas and methodology are easy to understand. There are some minor grammatical errors and awkward phrasings in places. Careful proofreading would improve readability and make the text smoother.
Author Response
Comments and Suggestions for Authors
Comments
The paper addresses lane-change trajectory planning for autonomous vehicles, which is an important and challenging problem. The main focus is investigation how to generate a smooth, feasible, and adaptive lane-change trajectory using model predictive control (MPC) that responds effectively to dynamic traffic conditions.
The topic is both original and highly relevant to the field of autonomous driving and intelligent vehicle control. Lane-change trajectory planning in dynamic scenarios is a critical challenge for autonomous vehicles, especially when considering real-time interactions with surrounding vehicles. While there has been prior research on trajectory planning, most existing studies focus on simplified or static traffic scenarios and do not fully address the complexities of dynamic, multi-vehicle interactions with sudden acceleration or deceleration events.
This study addresses a clear gap by proposing a method based on model predictive control (MPC) that integrates multi-objective optimization, real-time adaptability, and trajectory smoothing through coordinate conversion and polynomial fitting. The approach provides a systematic framework for generating safe, comfortable, and feasible lane-change trajectories in highly dynamic traffic conditions, which is not fully covered by current methods. Therefore, the work represents a meaningful contribution to advancing trajectory planning for autonomous vehicles in realistic highway scenarios.
This paper makes a meaningful contribution to the field of autonomous driving. Compared with previous work, it addresses lane-change trajectory planning in realistic dynamic scenarios, taking into account both preceding and left-behind vehicles and their sudden changes in speed. The method combines model predictive control with multi-objective optimization for safety, comfort, and maneuver speed, and uses coordinate conversion and polynomial fitting to generate smooth, feasible trajectories. The approach is validated under multiple traffic scenarios, showing good adaptability, which makes it more practical and closer to real-world highway conditions than many existing studies.
The methodology is generally sound, but several improvements could strengthen the study:
- Uncertainty and Sensor Noise: Consider incorporating uncertainties in vehicle behavior. This would make the MPC-based trajectory planning more robust to real-world conditions.
Answer: As suggested by the reviewer, comprehensively considering sensor accuracy and uncertainties in actual driving scenarios will enhance the robustness of the algorithm in practical applications. With the continuous development of sensor technology, their performance has become increasingly stable. This study was conducted under the assumption that sensors work efficiently. In the future, we will further optimize the algorithm by comprehensively considering sensor uncertainties.
- Parameter Sensitivity: Providing an analysis of how sensitive the method is to key parameters (e.g., weights in the objective function, prediction horizon) would help readers understand the method’s stability and reliability.
Answer: Yes, analyzing the sensitivity of the proposed method to key parameters (such as weights in the objective function and prediction horizon) will contribute to enhancing the stability and reliability of the method. In the future, we will further optimize the algorithm by comprehensively considering the sensitivity of various key parameters.
- The conclusions are consistent with the evidence and arguments presented, and they address the main research question. The paper clearly demonstrates, through simulation studies under multiple dynamic scenarios, that the proposed MPC-based lane-change trajectory planning method can generate smooth, safe, and feasible trajectories. The step-by-step methodology from kinematic modeling, multi-objective optimization, coordinate conversion, to polynomial fitting, is logically connected to the simulation results. The results show that the method adapts to changes in surrounding vehicles’ speeds, maintains safety distances, and achieves smooth lane changes, which directly supports the main question of how to plan lane changes in dynamic highway scenarios in real time. Therefore, the conclusions appropriately reflect the evidence and substantiate the proposed method’s effectiveness.
Answer: We really appreciate your comments.
- The references cited in the manuscript are general appropriate and relevant to the topic of autonomous vehicle trajectory planning and model predictive control (MPC). They cover a range of related areas, including MPC-based lane-change and collision avoidance, artificial potential field methods, sampling-based and optimization-based trajectory planning, reinforcement learning approaches, and kinematic modeling of vehicles. Most of the references are recent (2022–2025), which is good for showing the current state-of-the-art in the field. The inclusion of both journal articles and conference proceedings reflects the applied and experimental nature of trajectory planning research.
Answer: We really appreciate your comments.
- One minor point for improvement is that a few references are repeated (e.g., Yang et al., 2024, on MPC with adaptive APF appears twice), and the authors might consider removing duplicates. Overall, the references adequately support the methodology, discussion, and conclusions, and they demonstrate that the authors are aware of relevant prior work and research gaps. All figure are clear, well-labeled, and effectively support the methodology and results. No issues were observed, and they adequately illustrate the key points of the study.
Answer: Thank you very much for your suggestions. The duplicate references in the original manuscript have been removed and replaced, with red highlighting added for clarity.
Comments on the Quality of English Language
The English in the manuscript is generally clear. The ideas and methodology are easy to understand. There are some minor grammatical errors and awkward phrasings in places. Careful proofreading would improve readability and make the text smoother.
Answer: We really appreciate your comments.
Reviewer 2 Report
Comments and Suggestions for Authors
The paper focuses on the core issue of dynamic lane changing trajectory planning in autonomous driving, with a reasonable and practical method framework. The paper focuses on the core difficulty of "complex dynamic scene trajectory planning" in autonomous driving (dynamic randomness, high real-time requirements), and focuses on lane changing scenarios. Compared with static or simple constrained scenarios, it is closer to the real road environment (such as changes in surrounding vehicle speed), and its research value and application orientation are clear. Establish a kinematic model of the main vehicle and surrounding vehicles to provide a physical basis for prediction. Design multi-objective optimization functions and multiple constraints, and introduce relaxation variables to solve the feasibility problem of optimization, considering comprehensively. Combining least squares polynomial fitting to generate smooth trajectories, supplementing the engineering practicality of trajectory tracking (previewing the conversion of the following driver model to steering wheel angle), the method chain is complete. Based on MATLAB/Simulink and Prescan, a joint simulation platform was built. Three typical dynamic working conditions were selected, namely "constant speed of surrounding vehicles", "sudden braking of the front vehicle", and "sudden acceleration of the left rear vehicle". The adaptability of the method in dynamic scenes was preliminarily verified through indicators such as steering wheel angle, vehicle position, and relative distance. The simulation logic and scene coverage have certain rationality.
However, this article has shortcomings in highlighting innovative points and conducting comparative experiments
1. Insufficient depth of research review and unclear highlighting of innovative points
The evaluation of methods such as curve fitting, artificial potential field, sampling, and deep learning only lists their advantages and disadvantages, without in-depth analysis of their specific limitations in dynamic lane changing scenarios.
Although it is mentioned that the existing MPC methods have "high computational complexity" and "insufficient adaptability", the specific manifestations of these problems in dynamic lane changing are not clear, and the core differences between our method and existing MPC lane changing research are not clearly compared.
Suggest supplementing the specific limitations of existing methods in dynamic lane changing scenarios, adding a new section on "Innovation Points of this paper", and clarifying the differences with existing MPC methods.
2. Without comparative experiments, superiority cannot be demonstrated
Without comparison with existing trajectory planning methods such as traditional MPC, artificial potential field method, and deep learning method (such as lane changing time, lateral acceleration, and computation time), the advantages of our method cannot be proven; Insufficient coverage of working conditions (such as not considering lane changing on bends, multi vehicle interaction (with vehicles on both left and right lanes), and scenarios with different vehicle speeds (such as low-speed 40km/h and high-speed 100km/h), and the generalization has not been verified.
Add comparative experiments with traditional MPC and artificial potential field methods, and provide a comparison table of "lane changing time lateral acceleration calculation time"; Add new operating conditions (such as changing lanes on bends, multi vehicle interaction) to verify the generalization of the method.
Author Response
Comments and Suggestions for Authors
The paper focuses on the core issue of dynamic lane changing trajectory planning in autonomous driving, with a reasonable and practical method framework. The paper focuses on the core difficulty of "complex dynamic scene trajectory planning" in autonomous driving (dynamic randomness, high real-time requirements), and focuses on lane changing scenarios. Compared with static or simple constrained scenarios, it is closer to the real road environment (such as changes in surrounding vehicle speed), and its research value and application orientation are clear. Establish a kinematic model of the main vehicle and surrounding vehicles to provide a physical basis for prediction. Design multi-objective optimization functions and multiple constraints, and introduce relaxation variables to solve the feasibility problem of optimization, considering comprehensively. Combining least squares polynomial fitting to generate smooth trajectories, supplementing the engineering practicality of trajectory tracking (previewing the conversion of the following driver model to steering wheel angle), the method chain is complete. Based on MATLAB/Simulink and Prescan, a joint simulation platform was built. Three typical dynamic working conditions were selected, namely "constant speed of surrounding vehicles", "sudden braking of the front vehicle", and "sudden acceleration of the left rear vehicle". The adaptability of the method in dynamic scenes was preliminarily verified through indicators such as steering wheel angle, vehicle position, and relative distance. The simulation logic and scene coverage have certain rationality.
However, this article has shortcomings in highlighting innovative points and conducting comparative experiments
- Insufficient depth of research review and unclear highlighting of innovative points
The evaluation of methods such as curve fitting, artificial potential field, sampling, and deep learning only lists their advantages and disadvantages, without in-depth analysis of their specific limitations in dynamic lane changing scenarios.
Although it is mentioned that the existing MPC methods have "high computational complexity" and "insufficient adaptability", the specific manifestations of these problems in dynamic lane changing are not clear, and the core differences between our method and existing MPC lane changing research are not clearly compared.
Suggest supplementing the specific limitations of existing methods in dynamic lane changing scenarios, adding a new section on "Innovation Points of this paper", and clarifying the differences with existing MPC methods.
Answer: Thank you very much for your valuable comments. We have carefully revised the original manuscript in accordance with your comments: (1) A summary of the limitations of existing methods (such as curve fitting, artificial potential field, and sampling methods) in dynamic lane-changing scenarios has been added to the introduction section of the original manuscript. (2) A paragraph summarizing the innovations of this paper and its advantages over traditional methods has been supplemented in the original manuscript. All revisions have been marked with red highlighting in the original document.
- Without comparative experiments, superiority cannot be demonstrated
Without comparison with existing trajectory planning methods such as traditional MPC, artificial potential field method, and deep learning method (such as lane changing time, lateral acceleration, and computation time), the advantages of our method cannot be proven; Insufficient coverage of working conditions (such as not considering lane changing on bends, multi vehicle interaction (with vehicles on both left and right lanes), and scenarios with different vehicle speeds (such as low-speed 40km/h and high-speed 100km/h), and the generalization has not been verified.
Add comparative experiments with traditional MPC and artificial potential field methods, and provide a comparison table of "lane changing time lateral acceleration calculation time"; Add new operating conditions (such as changing lanes on bends, multi vehicle interaction) to verify the generalization of the method.
Answer: We really appreciate your comments. Existing trajectory planning methods such as the artificial potential field method and deep learning-based methods do have their unique characteristics, but they are mostly applied in the robotics field or the off-road low-speed autonomous vehicle domain. For the extreme condition of emergency collision avoidance, the rolling optimization feature of the MPC method is more in line with real driving requirements. Moreover, the iterative rolling optimization solution for dynamic driving scenarios is an advantage that the artificial potential field method and deep learning-based methods cannot match. Therefore, this study mainly conducts planning research based on MPC, and there is no need for additional comparative research with the artificial potential field method and deep learning-based methods. In addition, to further deepen the research content, future studies will be carried out under more complex working conditions, such as multi-vehicle interaction (involving vehicles on both left and right lanes) and scenarios with different vehicle speeds (e.g., low speed of 40 km/h and high speed of 100 km/h).
Reviewer 3 Report
Comments and Suggestions for Authors
The paper proposes a dynamic trajectory-planning method for autonomous vehicles using MPC. It focuses on lane-changing in complex dynamic scenarios, incorporating real-time adaptability via kinematic modelling, multi-objective optimisation, constraint handling, and least-squares curve fitting. I have some suggestions for the authors to consider,
- The paper’s novelty could be strengthened by explicitly contrasting the proposed MPC formulation with existing MPC variants. Add a table comparing computational complexity, constraint structure, and adaptability with previous studies.
- Discuss potential real-world implementation challenges, such as sensor delay or actuation constraints, to highlight the applicability of the approach.
- Consider adding robustness analysis under measurement noise or parameter uncertainty.
- Add more internationally recognised benchmark papers on similar work, such as 10.1109/TCYB.2022.3151880
- Include a brief discussion on how the proposed method could extend to multi-lane or curved-road environments.
Author Response
Comments and Suggestions for Authors
The paper proposes a dynamic trajectory-planning method for autonomous vehicles using MPC. It focuses on lane-changing in complex dynamic scenarios, incorporating real-time adaptability via kinematic modelling, multi-objective optimisation, constraint handling, and least-squares curve fitting. I have some suggestions for the authors to consider,
The paper’s novelty could be strengthened by explicitly contrasting the proposed MPC formulation with existing MPC variants. Add a table comparing computational complexity, constraint structure, and adaptability with previous studies.
Answer: We really appreciate your comments. This study focuses on the key issue of complex dynamic driving scenarios. Its core lies in leveraging the rolling optimization capability of MPC, which can fully consider historical information and operate under effective vehicle performance constraints. Based on the creatively established polar coordinate system, the study adopts an optimal iteration method, incorporates the steering, acceleration, and deceleration performance constraints of the host vehicle, and real-time searches for the optimal position of the vehicle at the next moment. By integrating the curve fitting method, it achieves adaptability to surrounding dynamic driving scenarios, thereby planning an ideal and feasible lane-changing trajectory for dynamic driving scenarios.
Discuss potential real-world implementation challenges, such as sensor delay or actuation constraints, to highlight the applicability of the approach.
Answer: We really appreciate your comments. With the continuous advancement of sensor technology, both its accuracy and stability have been steadily improved. This study was conducted under the assumption that sensors operate ideally. In the future, we will comprehensively consider potential real-world implementation challenges, such as sensor delay and actuation constraints, to further advance the optimization research of the algorithm.
Consider adding robustness analysis under measurement noise or parameter uncertainty.
Answer: We really appreciate your comments. Analyzing the sensitivity of this method to key parameters (such as weights in the objective function and prediction horizons) will help verify the stability and reliability of the method. The current main objective and innovation of this study lie in realizing trajectory planning that can adapt to variable environments under dynamic driving conditions, thereby achieving autonomous driving under full-range conditions. As suggested by the reviewers, in future work, we will consider adding robustness analysis under measurement noise or parameter uncertainty to further advance the optimization research of the algorithm.
Add more papers on similar work, such as 10.1109/TCYB.2022.3151880
Answer: Thank you very much for your valuable comments. We have carefully referred to the benchmark literatures you provided and optimized the content and format of this paper accordingly.
Include a brief discussion on how the proposed method could extend to multi-lane or curved-road environments.
Answer: Thank you very much for your valuable comments. The method proposed in this study is a dynamic trajectory planning approach. Its advantage lies in leveraging the rolling optimization characteristic of MPC to dynamically adapt to complex surrounding driving environments by iteratively searching for the optimal position at the next moment. Although the method description and simulation analysis in this paper are targeted at the typical scenario of a three-lane road with a preceding vehicle and a left-side vehicle, in more complex multi-lane or curved-road environments, the application in working conditions such as multi-lane, multi-vehicle, or curved-road scenarios can be achieved merely by designing reasonable corresponding constraints in the establishment of the multi-objective function.